# Distribution and Severity of Placental Lesions Caused by the *Chlamydia abortus* 1B Vaccine Strain in Vaccinated Ewes

**DOI:** 10.3390/pathogens10050543

**Published:** 2021-04-30

**Authors:** Sergio Gastón Caspe, Javier Palarea-Albaladejo, Clare Underwood, Morag Livingstone, Sean Ranjan Wattegedera, Elspeth Milne, Neil Donald Sargison, Francesca Chianini, David Longbottom

**Affiliations:** 1Moredun Research Institute, Edinburgh EH26 0PZ, UK; clare.underwood@moredun.ac.uk (C.U.); morag.livingstone@moredun.ac.uk (M.L.); sean.wattegedera@moredun.ac.uk (S.R.W.); francesca.chianini@moredun.ac.uk (F.C.); david.longbottom@moredun.ac.uk (D.L.); 2Royal (Dick) School of Veterinary Studies, University of Edinburgh, Edinburgh EH25 9RG, UK; elspeth.milne@ed.ac.uk (E.M.); neil.sargison@ed.ac.uk (N.D.S.); 3Estación Experimental Mercedes, Instituto Nacional de Tecnología Agropecuaria (INTA), Mercedes CP 3470, Argentina; 4Biomathematics and Statistics Scotland, Edinburgh EH9 3FDK, UK; javier.palarea@bioss.ac.uk

**Keywords:** enzootic abortion of ewes, *Chlamydia abortus* 1B vaccine strain, ovine placenta, principal component analysis, histology, immunohistochemistry

## Abstract

*Chlamydia abortus* infects livestock species worldwide and is the cause of enzootic abortion of ewes (EAE). In Europe, control of the disease is achieved using a live vaccine based on *C. abortus* 1B strain. Although the vaccine has been useful for controlling disease outbreaks, abortion events due to the vaccine have been reported. Recently, placental pathology resulting from a vaccine type strain (vt) infection has been reported and shown to be similar to that resulting from a natural wild-type (wt) infection. The aim of this study was to extend these observations by comparing the distribution and severity of the lesions, the composition of the predominating cell infiltrate, the amount of bacteria present and the role of the blood supply in infection. A novel system for grading the histological and pathological features present was developed and the resulting multi-parameter data were statistically transformed for exploration and visualisation through a tailored principal component analysis (PCA) to evaluate the difference between them. The analysis provided no evidence of meaningful differences between vt and wt strains in terms of the measured pathological parameters. The study also contributes a novel methodology for analysing the progression of infection in the placenta for other abortifacient pathogens.

## 1. Introduction

Enzootic abortion of ewes (EAE) is a complex disease that is caused by the obligate, intracellular Gram-negative bacterium *Chlamydia abortus* (*C. abortus*). In pregnant sheep infection can result in different clinical outcomes, namely abortion, the birth of weakly lambs, or the delivery of infected but apparently normal lambs [1]. Typically, EAE is characterised by severe necrosuppurative placentitis [1,2,3], typified by the presence of a variable number of placental lesions. The placenta and other products of lambing and abortion are the main sources for transmission of infection to other naïve animals [4,5].

Control of the disease is achieved in most European countries using the live *C. abortus* 1B vaccine strain [6]. This vaccine strain was developed by chemical mutagenesis of the *C. abortus* field strain AB7 and was believed to induce immunoprotection without causing disease because of its reduced growth at 39.5 °C in sheep [7]. However, this vaccine strain has been associated with sporadic abortion events in sheep [6,8,9]. Furthermore, a recent study [10] has demonstrated that lesions resulting from infection of the placentas of *C. abortus* 1B vaccinated animals are indistinguishable from those resulting from an experimental infection with wild-type (wt) *C. abortus* strain S26/3. However, we wanted to further explore the complexity of the infection that occurs by defining the changes involved in placental immunopathology due to the vaccine strain in comparison with a typical wt infection.

Thus, the purpose of this study was to characterise the distribution and severity of lesions in the placentas of sheep vaccinated with the 1B vaccine strain and wt strain S26/3 of *C. abortus* in different placental areas, from the surface to deeper underlying tissue layers and blood vessels. Specifically, the study was designed to investigate and assess a variety of histopathological and immunohistochemical parameters through a novel scoring system and implement a tailored multivariate statistical processing of these based on principal component analysis (PCA) [11]. The analysis enabled a detailed comparison and exploration of the relationships between the two strains in terms of the specific localisation of pathological features in the placental tissue. Ultimately, this will improve our understanding of the pathogenesis of the *C. abortus* 1B vaccine strain in pregnant vaccinated animals, while the approach taken may also be applicable to other reproductive disease models.

## 2. Materials and Methods

### 2.1. Ethics Statement

The two placentas used in the present study were collected at parturition from a flock of ewes on a working farm. These ewes had been purchased at market from an EAE-accredited Premium Health Scheme flock, vaccinated twice, two years apart, prior to mating with the commercial 1B vaccine (Cevac-*Chlamydia*, Ceva Animal Health Ltd., Amersham, UK). The vaccines were administered subcutaneously according to manufacturer’s instructions. As no experimental procedures, sampling or any other interventions were conducted on these animals, with only placentas being collected as part of normal lambing, this study was not subject to any UK Home Office licensing requirements.

Additionally, two wt (wt P1, *n* = 36 cotyledons; wt P2, *n* = 30) and two negative control (neg P1, *n* = 37 cotyledons; neg P2, *n* = 40) formalin-fixed placental samples from a previously conducted study [12] were used for comparative purposes. This study was carried out in strict accordance with the Animals (Scientific Procedures) Act 1986 and in compliance with all UK Home Office Inspectorate regulations, under the experimental protocol of the Moredun Experiments and Ethical Review Committee (Permit number: E30/17).

### 2.2. Scoring of Histopathological Features

The datasets and analyses described in this study were generated using placentas derived from *C. abortus* 1B strain-vaccinated ewes, as previously described in [10]. Briefly, two placentas presenting EAE compatible lesions, herein referred to as infected with a vaccine-type (vt) strain and designated vt-P1 and vt-P2, were analysed (vt P1, *n* = 46 cotyledons; vt P2, *n* = 51). Additionally, two placentas from ewes experimentally infected with wt *C. abortus* strain S26/3 (designated wt-P1 and wt-P2) and two additional placentas from EAE-free ewes, which lambed normally (designated neg-P1 and neg-P2), were used as negative controls. For each of these placentas, all cotyledons (C) and their surrounding intercotyledonary (IC) areas were trimmed and embedded in paraffin-wax. Serial sections 5 μm thick were cut from each C+IC block and mounted on glass microscope slides (Superfrost^®®^ Plus, Thermo Scientific, Braunschweig, Germany). Two sections for each cotyledon were analysed; one stained with haematoxylin eosin stain (HE) for histology and the other processed for immunohistochemical (IHC) analysis. After that, slides were scanned using a digital slide scanner (Nanozoomer XR, Hamamatsu, Photonics Ltd., Hertfordshire, UK) and the images analysed using pathology analysis software Qu-Path (https://qupath.github.io, accessed on 11 May 2020) [13].

For each placental area (C and IC), histopathological analysis was performed by layer, specifically cotyledonary trophoblast (CT) and intercotyledonary trophoblast (ICT) layers, and cotyledonary mesenchyme and intercotyledonary mesenchyme (ICM), examining 10 × 1 mm^2^ sections for each. In each area, a detailed evaluation was made of necrosis (N), polymorphonuclear leukocyte infiltration (PMNI), mononuclear infiltration (MI), vascular mural necrosis (VN), mononuclear perivascular infiltration (VMI), polymorphonuclear leukocyte perivascular infiltration (VPMNI), vascular endothelial activation (VEA) and vascular thrombosis (VT). Each of these parameters were scored according to the severity of the observed lesions on an ordinal scale of 0 to 5 (0, no significant lesions; 1, just perceptible; 2, mild; 3, moderate; 4, severe; and 5, very severe). Furthermore, each area examined was also classified according to its proximity to the main chorionic blood vessel (8.0–9.0 mm diameter) as proximal (Px: up to 3 cm distance) or distal (D: greater than 3 cm distance) to determine the relationship between the distribution of the lesions and the blood supply (Table 1).

### 2.3. Scoring for Immunolabelling in the Tissues

IHC was performed in the same placental areas detailed in Section 2.2 and in Table 1 using a genus-specific mouse monoclonal antibody raised against the lipopolysaccharide (LPS) of *C. abortus* strain S26/3 (mAb 13/4, Santa Cruz Biotechnology, Heidelberg, Germany), as previously described [14], but using goat anti-mouse IgG conjugate (Envision™ + System HRP labelled polymer, Dako, Ely, UK) with aminoethyl carbazole alcohol soluble chromogen (AEC, Vector Laboratories, Peterborough, UK) enzyme substrate for the labelling. The same 10 × 1 mm^2^ sections for each tissue area were analysed (Table 1). IHC labelling in the trophoblast and mesenchyme (TMIHC) or labelling in the blood vessels (VIHC), were scored on an ordinal scale from 0 to 4 (0, no labelling; 1, a single positive focus labelled in the observed field; 2, moderate labelling of two to four foci; 3, multifocal positive labelling of 5 to 9 foci in the observed field; 4, diffuse and/or very strong positive labelling in the observed field).

### 2.4. Statistical Analysis

The collected data consisted of scores in relation to a number of pathology parameters observed for the trophoblast and mesenchyme layer (N, PMNI, MI and TMIHC) and for blood vessels (VEA, VN, VMI, VPMNI, VT and VIHC) across areas (CM, CT, ICM and ICT) in a varied number of cotyledons (between 20 and 28 per sample) in different locations relative to the main blood vessel (D and Px) (Table 1). Datasets were generated using the parameters scored for placentas distributed into three treatment groups: group wt, *C. abortus* positive control (2 ewes; 1 placenta each); group vt, vaccinated animals (1 ewe; 2 placentas); and group neg, negative control (1 ewe, 2 placentas). The scoring generated 951 observations (363, 284 and 304 for groups wt, vt and neg, respectively). The scores given were ordinal values between 0 and 5 for all histological parameters and between 0 and 4 for IHC labelling.

An exploratory multivariate data analysis approach was applied to obtain an overall synthetic representation of the data and investigate potential patterns and associations between pathology parameters according to treatment groups. In particular, principal component analysis (PCA) is a statistical method widely used to examine variability and associative patterns in multivariate data sets [11]. It was used here to determine optimal linear combinations (principal components, PCs) of the pathology parameters that summarised the information (or variability) in the original data. The first PCs (PC1 and PC2) account for most of the original information and they were used to obtain a graphical representation in two dimensions called a biplot. In a biplot, both samples and pathology parameters are jointly represented by points and rays from the origin, respectively, which facilitates a visual assessment of relationships and comparisons at the expense of losing some information contained in the subsequent PCs. The proximity between the points represents the similarity between the samples according to the measured parameters. The rays indicate directions of increasing values of the parameters, with the origin in the biplot (point (0, 0)) corresponding to their average value in the original multi-parameter data set, and the angle between rays is proportional to the correlation between the corresponding parameters. However, standard PCA assumes ordinary numerical data, which was in contrast with the categorical ordinal nature of the score data for each pathology parameter. This was overcome by applying a so-called optimal scaling procedure [15], by which the ordinal variables were quantified into a continuous numerical scale making use of step functions in such a way that: (1) the variance accounted for in the quantified variables was maximised, and (2) the original order was respected (in this case scores from 0 to 4 or 5 according to severity). The resulting transformed (quantified) variables were then used as input in standard PCA. Moreover, the potential associations of the pathological parameters with location, either D or Px, were investigated by projecting these categories onto the PCA biplots as supplementary variables. Their coordinates in the biplot are determined by the barycentre of the samples that belong to each category. These data analyses were conducted separately for trophoblast and stroma, and blood vessel areas using the R system for statistical computing and the Gifi and factoMineR packages [16,17,18].

## 3. Results

### 3.1. Histopathological Analysis

Both vt placentas showed the typical multifocal necrosuppurative placentitis associated, in some cases, with vasculitis, mural necrosis of arteries and arterioles and thrombosis. The trophoblast layer was severely affected, being replaced by a densely compact layer composed mainly of an intense infiltration of polymorphonuclear leukocytes, cellular debris, and small number of mononuclear inflammatory cells (Figure 1A,C). Blood vessels showed a typical necrotising vasculitis with mixed inflammatory cell infiltration, associated in some cases with mural necrosis and thrombosis, principally in those located in the mesenchyme (Figure 1A,E and Figure 2A). The degree of occlusion of the arteries and arterioles was variable (Figure 1E and Figure 2A), but when it was completely or severely obstructed, there was severe ischemic necrosis of the surrounding tissues in the trophoblast layer irrigated by these vessels (Figure 1A and Figure 2C).

Similar histopathological lesions were found in the wt placentas. As was observed in vt placentas, wt placentas showed vasculitis (Figure 2B), thrombosis with different degrees of luminal occlusion associated (Figure 1B,F and Figure 2B,D) with severe necrosis of the trophoblast layer with (Figure 1B and Figure 2B) or without (Figure 2D) association with severe suppurative infiltration. No EAE-compatible lesions were observed in cotyledon and intercotyledonary areas from the negative control placentas (neg-P1 and neg-P2).

### 3.2. Immunohistochemical Labelling

Immunohistochemical labelling using the *C. abortus* genus-specific LPS revealed an intense positivity with a moderate to multifocal distribution in the majority of the cotyledons in both vt and wt placentas. The most intense TMIHC labelling was observed in the trophoblast layer showing a predominant score of between 2 and 4 (Figure 3E,F), with minimal or null labelling in the mesenchyme. On the other hand, labelling in blood vessels (VIHC) was negative in most of the sections, with the exception of 4 arterioles (one in vt-P2, one in wt-P1 and two in wt-P2) which showed single focal positive labelling in the tunica intima. No positive IHC labelling was observed in any cotyledon or intercotyledonary areas from the negative control placentas (neg-P1 and neg-P2).

### 3.3. Statistical Analysis

Figure 4 illustrates summary statistics of the score data (1st quartile, median and third quartile, which are compatible with their ordinal nature) for each pathological parameter by area, location, and treatment group. As is evident by the increase in average scores, indicated by a change in colour gradation from green through to red (scores 0 to 4 or 5), there is a clear distinction between what was observed in the chorionic epithelium (trophoblast) and mesenchyme compared to the blood vessels, with the first showing higher scores overall. For trophoblast (CT; ICT) and mesenchyme (CM; ICM), differences in median value (Med) scores for N and TMIHC parameters were generally higher in trophoblast, although some scores were generally similar across the three groups (wt, vt and neg). For example, for CT the three groups were found to have similar scores for each parameter except TMIHC, while greater differentiation in scores between neg and infected (vt/wt) was observed in ICT for N, PMNI and TMIHC parameters. In mesenchyme, scores were similar among groups, with the main exceptions being for TMIHC (in CM and ICM) and for N (in ICM). For the inflammatory infiltration parameters, PMNI and MI, no clear patterns were observed differentiating wt/vt and neg groups. For blood vessels, the perivascular infiltration and vasculitis parameters (VMI, VPMNI) and thrombosis (VT) show a distinctive pattern in relation to area, showing higher scores in the mesenchyme layer for all groups. For VIHC, no difference was observed neither among groups nor areas. No evident differences in average scores according to the location (D and Px) were observed.

The PCA-based biplot diagrams using the first two principal components (PC1 and PC2) were produced for the chorionic epithelium and mesenchyme (Figure 5) and for the blood vessels (Figure 6), and for each of the placental areas within them, either cotyledonary (CT, CM) or intercotyledonary (ICT, ICM). In relation to trophoblast and mesenchyme layers for both areas, the cotyledonary and intercotyledonary area (Figure 5), the fractions of original data variability accounted for by the biplots ranged from an acceptable 79–85.7%. The biplots show a comparable overall configuration of points, and hence, there appears to be no relevant differences according to the area (cotyledonary (CT, CM) and intercotyledonary (ICT, ICM)). The ray for N is very much parallel to the PC1 axis, followed by the rays for PMNI and MI. Hence, these parameters are mostly associated with PC1 and not with PC2. They are also relatively close to each other, especially in the cases of ICT and ICM, thus reflecting a marked correlation between them. On the other hand, TMIHC is more associated with PC2 and appears fairly perpendicular to the others in the graph (particularly to PMNI and MI), hence indicating weak to very weak correlation with them. The treatment groups are principally distinguished along the direction of the TMIHC ray, i.e., according to the scores given to the samples in this parameter, with the lowest TMIHC scores being observed in group neg and the highest in group vt (the larger symbol within each group represents the respective mean value). The largest spread in scores is observed in groups vt and wt and most overlapping occurs between samples in these two groups. The neg group is distinguished at the bottom-left corner, i.e., most samples in this group are placed within the region of low scores in all pathology parameters.

PCA was also conducted on the pathology parameter variables for the blood vessels. Five markedly outlying samples were identified in a preliminary examination, particularly in relation to VIHC scores. Eliminating these cases produced improved results and increased the fraction of the total data variability explained by the associated biplots (Figure 6). The fractions of data variability accounted for by the biplots ranged from 64.2–75.1%, which, although still accounting for a reasonable amount of total variability, implies that a larger portion of the information is lost in the biplot representation of the data when compared with trophoblast and mesenchyme (Figure 5). The overall configuration is again comparable across areas (C and IC), with VPMNI, VMI, VT and VN, generally linked to PC1 and exhibiting moderate to high positive correlations between them. The VEA ray is generally separate from the others, particularly for areas CT and ICT (Figure 6A,B) and in relation to parameters VN and VT, thus having the weakest correlations with them.

It was observed that VIHC has a very short ray that is difficult to distinguish from the origin, and thus, very little information about VIHC is represented in the biplot. This is explained by the fact that almost all the results were negative for VIHC. Although most parameter scores appear variable along the horizontal axis (with increasing scores to the right-hand side) and, hence, mostly associated with PC1 (particularly VPMNI), no strong association is observed between any of them (suggested by the low to moderate proximity between rays), except perhaps between VN and VT, which are the closest ones. In terms of treatment groups, although some ordering can be distinguished from left (group neg) to right (group wt) (e.g., for ICM; Figure 6D), the dispersion of the points is very high and there is no evident separation between the respective samples according to pathological scores across areas as seen for trophoblast and mesenchyme above, between the neg and the other two groups (Figure 4), with a distinction being only more evident in the case of ICM (which is the biplot explaining the highest fraction of the original variability, 75.1%).

For the parameter distal (D) and proximal (Px) locations, they were initially added as supplementary variables to the PCA solution and associated biplots. However, they were placed very close to the origin of the biplot display and were hardly distinguishable, thus indicating a very weak relationship with the PCs and associated parameters. For this reason, they were then omitted from Figure 4 and Figure 5 to simplify visualisation.

## 4. Discussion

This study describes detailed localisation, histopathological and statistical analyses of lesions found in two placentas that resulted from a sheep that lambed normally and that had been vaccinated with the commercial live *C. abortus* vaccine as part of a previously published study [10]. The study showed that both of these placentas, referred to as vt-P1 and vt-P2, exhibited differences in their extent of gross pathology, with vt-P1 showing lesions across 60% of the placental surface and vt-P2 showing around 15%. Despite these differences, which are characteristic observations for this disease [2,5,19], histopathology and immunohistochemical analysis for the presence of chlamydial LPS antigen revealed no differences with lesions observed following experimental infection with a typical wt field strain.

In this study, we undertook a full histological examination of the placental cotyledons and intercotyledonary areas from the vt and wt placentas and adopted a novel system for grading the histological and pathological features present in the tissues. This was accompanied by a tailored statistical processing based on PCA and a biplot graphical display that allowed for a global assessment and visualisation of the differences between them based on multiple pathology parameter scores. Specific multivariate methods such as PCA, which reduce the dimension of the original dataset while preserving as much information as possible and accounting for the structure of associations in the data, were shown to be helpful for the purpose in a multi-parameter context as a more insightful alternative than simplistic analyses based on individual parameters in isolation. This simultaneous analysis facilitated meaningful comparisons between treatment groups and the identification of interactions between pathological parameters that allowed insights into prospective relationships.

When comparing the lesions observed in vt and wt placentas in the trophoblast and mesenchyme layers of the placental tissue, the data variability accounted for by the biplots ranged from 79–85.7%, meaning that the multi-parameter data were satisfactorily represented in two dimensions. In addition, the distribution of the points in the graphs presented a considerable overlap between group vt and wt but was highly distinguishable from group neg, indicating that the lesions associated with vt-strain infection are similar to those produced by wt-strains, further supporting our previous observations [10]. In this study, the results of the histopathological and PCA biplots of the placentas show that there are only minor differences in the severity and distribution of the lesions observed in the vt-placentas and in wt-strains infections, suggesting that the pathogenesis associated with the infection of any of these both strains follow similar mechanisms. Moreover, the configuration of scores and associations between pathology parameters were comparable across areas.

Histologically, although both vt and wt placentas show a typical EAE necrosuppurative placentitis, the analysis was extended to investigate whether there was any association between the lesions and their proximity to the large chorionic vessels, denoted as proximal (Px) and distal (D). On examination of the results of this study, it was evident that D and Px locations were very poorly associated with the pathological parameters. In addition, the highest scores for necrosis and TMIHC in this study were principally localised to the trophoblast layer for the inoculated groups in both CT and ITC, showing the important role of the trophoblast in the spread of the infection. In fact, these findings provided evidence that that the distribution and severity of the pathogenesis associated with infection might not be directly related to the main blood vessel, but rather the anatomic proximity to the initial infected cotyledon. This fits with the current view that transmission of infection occurs following the development of hematomas at the bases of the chorionic villi as a result of leakage from peripheral capillaries in the maternal septal tips. This allows the infection of placental chorionic epithelial (trophoblast) and the spread of infection centrifugally to neighbouring cells and the peri-placentome membranes [1,20]. Therefore, it is probable that the cotyledons can be infected simultaneously or in a random manner, as shown by the variable distribution of necrosis of the trophoblast cells and oedema in the mesenchyme and of the presence of lesions.

Analysis of the histopathological findings from the biplots revealed a marked correlation between N, PMNI and MI, especially in the intercotyledonary area, and presented a lot of overlap between the vt and wt groups. However, IHC labelling for *C. abortus* in the trophoblast and mesenchyme (TMIHC) appeared weakly correlated with the other parameters, especially with regards to inflammatory infiltration (PMNI and MI). This may be as expected because the function of the inflammatory cells, which is to eliminate the microorganism, may also cause necrosis in the process [2,20]. Therefore, it would be logical to assume that when cell infiltration is acute and mild, there is a greater chance of bacteria being detected than when the inflammatory infiltration and necrosis are more severe, denoting a more chronic lesion. For the obligate intracellular pathogen *C. abortus*, this is likely to occur because it requires cells to survive and replicate. Therefore, when necrosis is diffuse and severe, the EBs are forced to move to the surrounding areas where there are still viable chorionic epithelial cells to be infected. Remarkably, TMIHC scores for the strains evaluated showed high levels of overlap, meaning that both exhibit a similar pattern of chlamydial antigen distribution, suggesting a similarity in the way infection spreads.

In blood vessels, the observations in neg group placentas were far more disperse and also there was a greater overlap between observations in vt and wt placentas. In relation to the areas, analysis revealed a tendency towards a higher score for inflammatory infiltration (VMI, VPMNI) and thrombosis (VT) in the blood vessels in the deeper mesenchyme (CM and ICM) rather than trophoblast layer (CT, ICT) in the vt and wt placentas, perhaps not surprising given their anatomical location, function, and the abundant presence of blood vessels in the mesenchyme. However, overall, no strong associations between pathology parameters were observed as suggested by the low to moderate proximity observed in biplots. Although no associations were evident, the higher recruitment of inflammatory cells and also the associated inflammatory reaction in the mesenchyme observed in vt and wt infections could be related to the PMN infiltration, cellular necrosis, lysis of the bacteria in the trophoblast layer that produces chemotaxis of these cells and also the thrombosis that is associated with the chlamydial LPS, all of which are processes associated with infection. As for links with location, the conclusions are analogous to what was observed for the epithelium/stroma, showing no relevant associations with the proximity of the major placental vessels. Interestingly, VIHC was found to essentially play no role in data variability. That is because when comparing VIHC scores, which were analysed according to group, almost all observations had a score of zero. This indicates that the chlamydial LPS antigen is virtually absent from the blood vessel based on negative IHC labelling, supporting the previous observations about the distribution of chlamydial antigen.

## 5. Conclusions

This study analysed the distribution and severity of pathology resulting from infection of the ovine placenta with the *C. abortus* live 1B vaccine strain in comparison to that resulting from a typical wt strain. This was achieved through the development and application of a novel grading system based on histopathological and immunohistochemical analyses and the scoring of a range of pathological parameters. PCA was used as an objective analytical tool in order to compare these features simultaneously. This resultant analysis provides further detailed evidence that the commercial live vaccine strain can cause EAE infections that are indistinguishable from those observed in the field, and thus, expands our knowledge on the pathogenesis and role of the vaccine strain in causing disease. Furthermore, the approach taken may have applicability to other reproductive pathogens and future studies on placental pathology and pregnancy outcome. Despite previous evidence that the commercial live vaccine strain can cause infection and disease in some animals [6,8,9,21] and the further evidence presented here supporting this, this vaccine has nonetheless had an important role in controlling EAE for the last 25 years since its release in the mid-1990s. For this reason, we anticipate that the vaccine will continue to be used until such time that a new next generation vaccine that is safer and unable to cause disease is developed.

## Figures and Tables

**Figure 1 pathogens-10-00543-f001:**
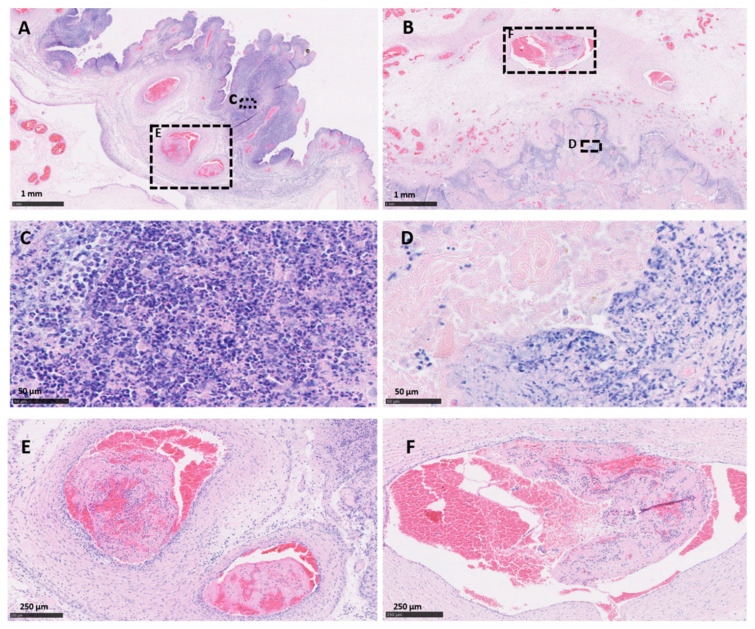
Histopathological changes in the placentas of sheep infected with *C. abortus* vaccine-type and wild-type strains. (**A**,**C**,**E**) Placenta vt-P1: showing (**A**) necrosuppurative placentitis (expanded in (**C**)) associated with partially occlusive thrombosis (expanded in (**E**)); (**B**,**D**,**F**) Placenta wt-P2: necrosuppurative placentitis (**B**) showing fibrinoid necrosis of the trophoblast cells with debris cellular material and degenerated inflammatory cells (expanded in (**D**)), and partially occlusive thrombosis of the artery (expanded in (**F**)). Haematoxylin and eosin. Scale bars are as indicated.

**Figure 2 pathogens-10-00543-f002:**
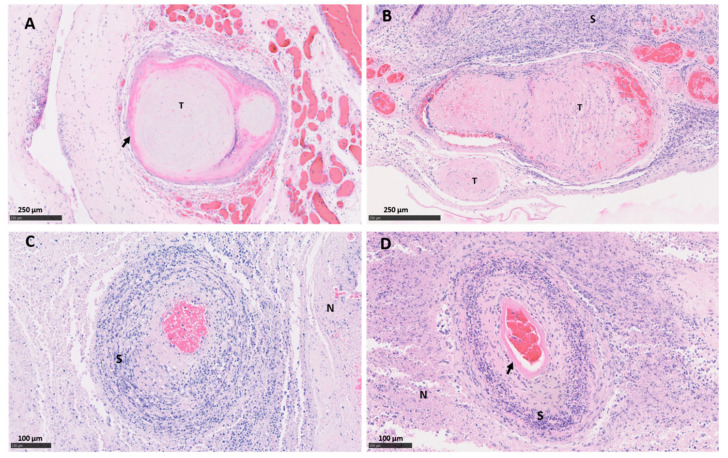
Histopathological changes in the placental blood vessels of sheep infected with *C. abortus* vaccine-type and wild-type strains. (**A**) Placenta vt-P1: showing occlusive fibrinoid thrombosis (T) with revascularisation in multiple small surrounding vessels; (**B**) Placenta wt-P1: showing severe thrombosis (T) with severe fibrinoid necrotising arteritis (arrow) and a dense band of amorphous and intensely leucocytic perivascular infiltrate (S); (**C**) Placenta vt-P2: necrosuppurative vasculitis, with severe mixed inflammatory infiltration compact layer (S) and necrosis of the surrounding tissue (N); (**D**) Placenta wt-P2, severe necrotising arteritis with mural necrosis (arrow), perivascular severe mixed inflammatory infiltration and ischemic necrosis of the surrounding tissue (N). Haematoxylin and eosin. Scale bars are as indicated.

**Figure 3 pathogens-10-00543-f003:**
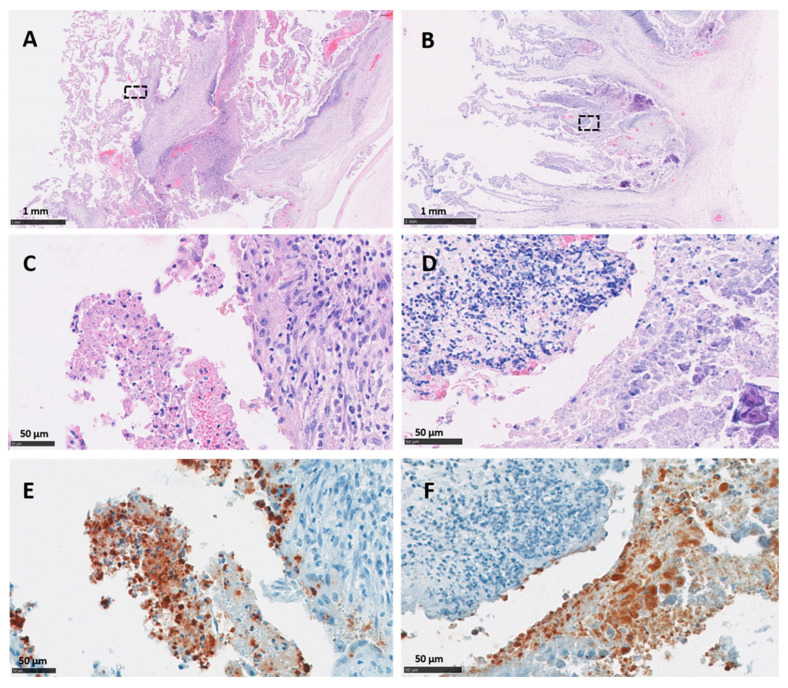
Histopathological changes and IHC labelling in the placentas of sheep infected with *C. abortus* vaccine-type strains. (**A**,**C**,**E**) Placenta vt-P1: showing (**A**,**C**) necrosuppurative placentitis with a large numbers of leucocytes in the chorionic epithelium of the cotyledon attached to superficial amorphous necrotic material stained by HE technique; (**E**) chlamydial organisms labelled by IHC using anti-LPS mAb 13/4 and counterstained with haematoxylin; (**B**,**D**,**F**) Placenta vt-P2: showing (**B**,**D**) necrosuppurative placentitis and extensive compact layer of leucocytes in the trophoblast layer and basement membrane stained by HE technique and (**F**) chlamydial organisms labelled by IHC using anti-LPS mAb 13/4 and counterstained with haematoxylin. Black outlined areas shown in top images indicates areas expanded in images located immediately below. Scale bars are as indicated.

**Figure 4 pathogens-10-00543-f004:**
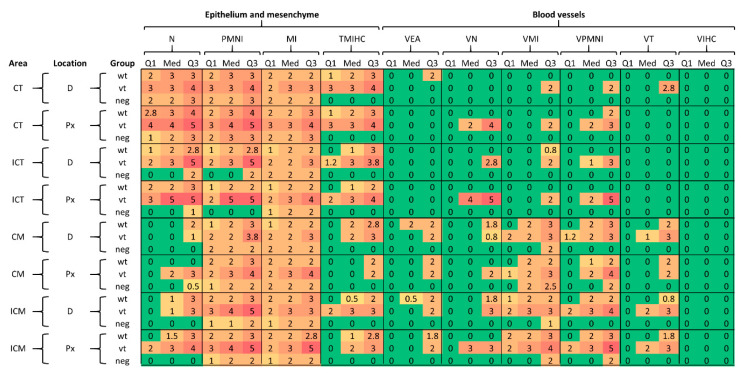
Summary statistics of the score data by area, location and treatment group for wild type (wt), vaccine type (vt) and negative control (neg) samples. Abbreviations for pathological parameters and placental location/area are defined in Table 1. Q1, Med and Q3 indicate first quartile, median and third quartiles, respectively.

**Figure 5 pathogens-10-00543-f005:**
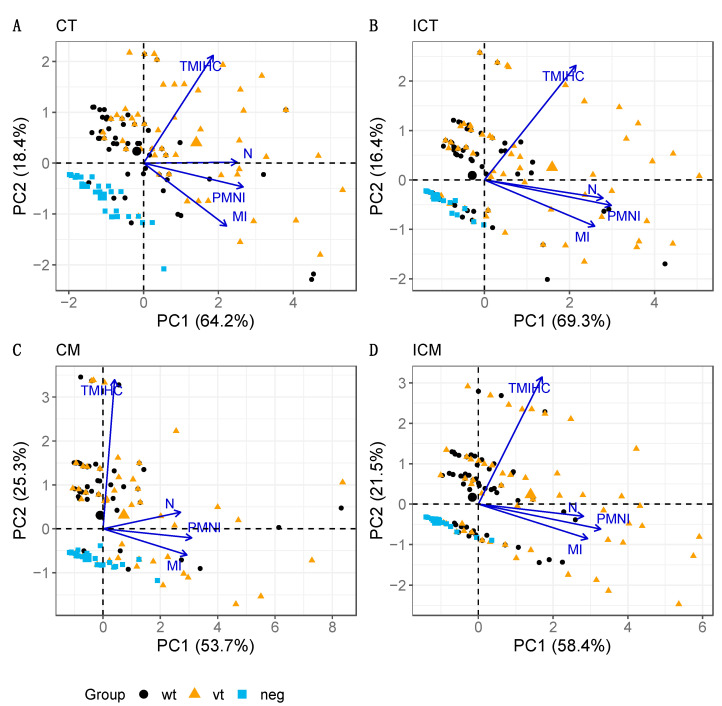
PCA biplots for trophoblast and mesenchyme layers in cotyledonary (**A**: CT, **C**: CM) and intercotyledonary (**B**: ICT, **D**: ICM) areas. The parameters analysed are necrosis (N), infiltration of polymorphonuclear leukocytes (PMNI), infiltration of mononuclear cells (MI), and labelling of chlamydial LPS by IHC (TMIHC). Note that the four biplots show very comparable patterns in terms of both, configuration of the treatment group points and associations between pathology parameters. The percentage of explained data variability by each PC axis is indicated in parenthesis.

**Figure 6 pathogens-10-00543-f006:**
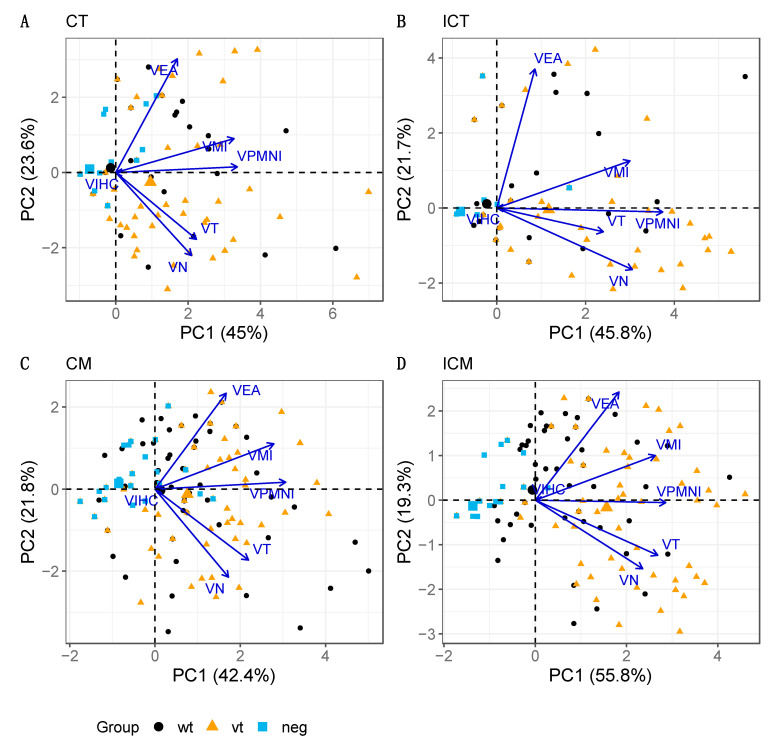
PCA biplot for blood vessels in cotyledonary (**A**: CT, **C**: CM) and intercotyledonary (**B**: ICT, **D**: ICM) areas. The parameters analysed are mural necrosis (VN), vasculitis-infiltration of polymorphonuclear leukocytes (VPMNI), vasculitis-infiltration of mononuclear cells (VMI), vascular endothelial activation (VEA), vascular thrombosis (VT) and labelling by IHC for *C. abortus* LPS (VIHC). Note that the four biplots show very comparable patterns in terms of both configuration of the treatment group points and associations between pathology parameters. The percentage of explained data variability by each PC axis is indicated in parenthesis.

**Table 1 pathogens-10-00543-t001:** Placental areas and pathological parameters analysed.

Location ^a^	Placental Area ^b^		Tissue Staining and Labelling ^c^			
	Cotyledon (C)	Intercotyledonary Membrane (IC)	Epithelium and Stroma		Blood Vessels ^d^	
			Histopathology Features	IHC	Histopathology Features	IHC
Proximal (Px)	Trophoblast (CT)	Chorionic trophoblast (ICT)	Necrosis (N)	Trophoblast and mesenchyme (TMIHC)	Vascular mural necrosis (VN)	Blood vessels (VIHC)
Distal (D)	Mesenchyme	Mesenchyme (ICM)	Polymorphonuclear leukocyte infiltration (PMNI) Mononuclear infiltration (MI)		Polymorphonuclear leukocyte perivascular infiltration (VPMNI) Mononuclear perivascular inflammatory infiltration (VMI) Vascular thrombosis (VT) Vascular endothelial activation (VEA)	

^a^ Location of analysed areas in relation to the main chorionic blood vessel. ^b^ Placental areas and tissue layers analysed in each location. ^c^ Histopathological features and IHC staining analysed in each tissue type (epithelium/stroma and blood vessels) in each tissue layer. ^d^ Blood vessels were analysed for each tissue layer to determine where the lesions are more prominent. The abbreviations for each pathological feature (in parentheses) are used throughout this article.

## Data Availability

Not applicable.

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
