# Peer review of "Distribution and Severity of Placental Lesions Caused by the *Chlamydia abortus* 1B Vaccine Strain in Vaccinated Ewes"

_pathogens, 2021, doi:10.3390/pathogens10050543_

Round 1

Reviewer 1 Report

In this manuscript Caspe et al. compared the placental pathology in ewes infected with a vaccine based on the 1B strain, versus a wild type strain (S26/3) of Chlamydia abortus. The main parameters analyzed were, the distribution and severity of the lesions, the composition of the cell infiltrates, the amount of bacteria and the role of the blood supply in infection. The tissues were stained with hematoxilin and eosin and a monoclonal antibody to the LPS of C. abortus S26/3. The authors utilized a newly developed system for grading the histopathological lesions and the multi-parameter data was statistically transformed through a tailored principal component analysis to identify the differences between the two sets of tissues. Caspe et al. concluded that there was not significant differences in the histopathological changes found in the placentas from ewes infected with the vaccine versus the wild type strain of C. abortus. In a previous study by this group, using standard diagnostic procedures, the authors reached the same conclusions.   

Strenghts of the Study

This new diagnostic approach may help analyze and compare pathological changes in various tissues infected with the same or different pathogens.

Limitations of the study:

  • This study only confirms the results previously reported by the same group of investigators (Caspe et al. PlosOne, November 16, 2020).
  • Only two placentas, from a single ewe, infected with the vaccine strain were studied.
  • In some of the Figures different magnifications were used to display images of tissues from the vaccine versus the wild type C. abortus infected ewes.

Author Response

We thank the reviewer for their comments. While we acknowledge the limited nature of the investigation being on two placentas, this was the result of the biological nature of the infection due to the vaccine strain in the 117 placentas analysed and thus the outcome was beyond our control. This was one of the reasons we wished to expand the analysis in the previous publication by developing and conducting a much more rigorous and detailed analysis of different immunological parameters and the relationship between these parameters to help understand the exact nature of the infection compared to a normal wild-type infection. An additional outcome of this analysis was that we have defined an approach and a robust statistical methodology that can be applied to other investigations on this and other abortifacient pathogens.

The comment regarding magnification in the figures has been dealt with in the revised Figures 1, 2 & 3 to ensure that the same magnification is used for both vaccine and wild-type images.

Reviewer 2 Report

The paper by Caspe et al. is a follow-up study of the same authors published in 2020 (PLosOne). The authors compare the patho-histological changes in placentas after OEA induced by the vaccine strain and the wildtype strain. In this study, they etablished a novel grading system including HE and IHC findings. I cannot comment much abot the PCA but more on the histological analysis. The findings confirm the results of the previous study. Everything is covered in the discussion section. There are no major issues. The only two minor issues refer to the Figures:

  • Figure 2D: The right vessel looks more like as it would contain some calcified material but not a thrombus
  • Figure 3D: this is not a suitable representation of "compact layer of leucocytes in the trophoblast layer and the BM". Please select another area and expand it 

Author Response

We thank the reviewer for their comments, which are addressed as follows.

Figure 2D: On reflection, we agree that it is not the best photograph for showing a typical thrombosis. The image has been replaced to illustrate a better representation in 2B and 2D was replaced with a similar image to 2C.

Figure 3D: Another area has been selected and the image replaced, as well as the corresponding IHC labelled image in Figure 3F.

Reviewer 3 Report

This study described novel use of PCA analyses for pathology descriptions and associations in C. abortus ovine abortions caused by vaccine and/or wt strains. Furthermore, the authors developed scoring system for pathology and grade of severity. I only have minor comments for authors. In M and M section 2.2 and 2.1 Lines 68 and 85: it is not clear what was the number of examined placentas (or how many in each dataset) were used for analyses. Same for control placentas. This needs to be stated exactly. Similar comment for Line 96: how many slides were examined per each tissue section? Finally, if wt and vaccine strain can produce same pathology, does this means that the vaccine should no be used anymore for C. ab infection control? A comment in discussion perhaps could be made.

Author Response

We thank reviewer #1 for their comments, which are addressed as follows.

..."M and M section 2.2 and 2.1 Lines 68 and 85"... The specific details on the number of placentas and number of cotyledons examined per placenta have been added on lines 68, 76-77 and 87-88 in the revised manuscript “pathogens-1189942_Manuscript with tracked changes”.

..."Similar comment for Line 96, number of slides examined"... This was an omission on our part. This has now been rectified on lines 95-97 in the revised manuscript “pathogens-1189942_Manuscript with tracked changes”.

..."The vaccine should no be used anymore for C. ab infection control?"...: While we understand the point made by the reviewer, we do not think that the vaccine should be withdrawn at this stage. However, we have added a statement to the Conclusion rather than the Discussion regarding the implications of this and other studies, as well as the need for the next generation of vaccine that is safer (lines 439-444). This change has resulted in the addition of an extra reference (#21).